# Medical students' attitudes toward psychiatry in Indonesia

**Fransiska Kaligis** [1,2]☯*, **Ribka Hillary**[3]☯, **Nabilla Merdika Putri Kusuma**[3]☯, **Helisa Rachel Patricie Sianipar**[3]☯, **Camilla Sophi Ramadhanti**[3]☯, **Ardi Findyartini**[4]☯, **Madhyra Tri Indraswari**[1‡], **Clarissa Cita Magdalena**[1‡], **Garda Widhi Nurraga**[1‡]

**1** Faculty of Medicine Universitas Indonesia, Department of Psychiatry, Jakarta, Indonesia, **2** Cipto Mangunkusumo National General Hospital, Jakarta, Indonesia, **3** Faculty of Medicine Universitas Indonesia, Jakarta, Indonesia, **4** Medical Education Unit, Faculty of Medicine Universitas Indonesia, Jakarta, Indonesia

☯ These authors contributed equally to this work.
‡ These authors also contributed equally to this work.
* fransiska.kaligis@ui.ac.id

**Data Availability Statement:** The SPSS files used in this paper has been uploaded in the database Open Science Framework: osf.io/w9eba.

## Abstract

Nearly two-thirds of psychiatric patients are reluctant to seek help from healthcare professionals due to stigma, discrimination, and negligence that evolve around the community, including healthcare providers. Future health professionals should have a positive attitude toward psychiatry and patients with mental health problems. Thus, it is vital to identify medical students' attitudes toward psychiatry as future healthcare providers. The authors conducted a cross-sectional study by using online questionnaires of "Perceptions of Psychiatry" in Faculty of Medicine Universitas Indonesia, with first- and fourth-year students (before psychiatric rotation), as well as fifth-year students and alumni (after psychiatric rotation). Out of 250 questionnaires distributed, 224 subjects responded, with a response rate 89.6%. *Chi-square* or *fisher* analysis was conducted to know the correlation between gender and attitudes towards psychiatry. The frequency distribution method was applied to identify the degree of stigmatization from respondents. A mix of positive and negative perceptions towards psychiatry was identified. The overall response was favorable to both before and after psychiatric rotation groups. Differences in perceptions between male and female psychiatry students as a discipline and career were statistically significant. Correcting misapprehension and removing the stigma on psychiatry during medical education might decrease the stigma in the psychiatric field and patients.

## Introduction

According to the World Health Organization's report, mental disorders are the leading cause of disability for people aged 15–44, accounting for nearly 40 percent of all disabilities in the same age group. It was also reported that the most common causes of years lived with disability are neuropsychiatric disorders, such as depression, alcohol-use disorders, schizophrenia, and bipolar disorder [1].

Over the past decades, public opinion about psychiatry has been generally negative, proven by the ongoing stigma towards psychiatric patients. Our society still owns the assumption that

**Funding:** The authors received no specific funding for this work.

**Competing interests:** The authors have declared that no competing interests exist.

psychiatric patients are dangerous, and the diagnosis and medical treatment for mental illness are unpredictable, resulting in distress among psychiatric patients due to stigma [2].

Mental health problem is a severe issue in Indonesia. People with mental health problems struggle to seek help, yet they must be stigmatized. In the community, the lack of knowledge regarding mental health condition is correlated with stigma. The developing stigma resulted in more than half of psychiatric patients in Indonesia being unwilling to be treated by healthcare professionals [3]. The stigma could even be found among healthcare professionals. According to Lim et al. [4], psychiatric patients were unwilling to go to health facilities due to fear of the mental illness stigma they would receive from society. Factors influencing this stigma in society may be related to the lack of knowledge and information regarding the course of mental illness and its treatment [3].

Stigmatization towards psychiatric patients mostly occurs to those with either schizophrenia or people with substance use problems [5]. There has been a growing concern that psychiatric patients are dangerous, unpredictable, and challenging to communicate. Over time, they are marked as the stereotype for all patients with mental illness and mental health condition, regardless of the underlying cause.

A 2010 study by Sartorius et al. [6] also found the perception that at times psychiatrists were not considered as other medical doctors and were often underestimated. The study also suggests that the subjects could not distinguish between a psychologist and a psychiatrist. At the same time, only a few of them answered that psychiatrists had the competency to prescribe medications.

The public's perspectives toward psychiatry have been primarily formed by the portrayal in the media, culture, and history. These images have also influenced how medical students are being trained. For example, it was found that doctors would criticize how psychiatry lacks evidence-based theories and demands emotions more than scientific understanding, as well as criticism of how psychiatrists are not apt to be role models [7,8].

Medical students were reported to have minimal knowledge in psychiatry with a large amount of stigma prior to being exposed to psychiatric patients [8]. However, the same study also suggested that stigma decreases as students enter psychiatry clinical rotation and meet psychiatric patients [8]. Another study in 2005 by Bulbena et al. [9] found that the percentage of students who were willing to choose psychiatry as their specialty increased to 10.4% after the exposure to psychiatric rotation. The factors contributing to these findings are subjective to the clinical rotation and other factors unrelated to psychiatric training. Cutler et al. [10] showed that the preference of medical students in choosing psychiatry as a career was often affected by their family members. Students with whom families with negative perspectives toward psychiatry surround would less likely choose psychiatry as their future career.

In Indonesia, no research has been conducted about how the negative stigma is circulating and how it affects medical students' future decisions and preferences.

This research was conducted by involving medical students of Faculty of Medicine Universitas Indonesia to identify the view of medical students towards psychiatry as a discipline and career choice, psychiatric training, treatments, patients, and profession of psychiatrist as a role model. We also aimed to see whether the attitude may differ with the duration of education and exposure to psychiatric rotation. The medical students' view of psychiatry is crucial in ensuring equal management for patients as they will soon become medical professionals. Therefore, identifying the attitudes of health professionals starting from their early stages can help plan further approaches to raise awareness of the problem and improve stigma towards psychiatry.

## Material and methods

The cross-sectional method was used in this research using online questionnaires, running for over six months from May 2019 in Faculty of Medicine Universitas Indonesia. The study was approved by Ethics Committee of Faculty of Medicine Universitas Indonesia, number 900/ UN2.F1/ETIK/V/2019. All subjects provided written consent before giving their responses, which the assessed data in the study were shared publicly without interrupting participants privacy.

A simple random sampling method was used to acquire samples. The subjects were randomly selected from a list of student ID numbers. The inclusion criteria were: (1) first-year medical students before any exposure to psychiatry module, (2) fourth-year medical students who have passed pre-clinical psychiatry module but prior to clinical psychiatric rotation, (3) fifth-year medical students after the exposure to psychiatric rotation, (4) fresh graduate alumni who were in the one-year mandatory medical service assignment (Internship). In Universitas Indonesia, pre-clinical neuro-psychiatry module is given in the last two months of the third year, whereas the psychiatry clinical rotation is done in the fifth year.

The "Perceptions of Psychiatry" questionnaire used in this study, consisted of 27 questions self-assessment survey, which was adapted and developed from Balon et al. [11] in 1999 and had been used in a similar survey conducted by Stuart et al. [7]. The listed items in the questionnaire would measure the perceptions of psychiatry as a discipline (5 items), psychiatric treatment (7 items), psychiatrists as role models (5 items), psychiatry as a career (7 items), psychiatric patients (7 items), and psychiatry training (5 items). The scoring for each question were based on Likert's scaling method of four options; strongly agree, agree, disagree, and strongly disagree. The questions were arranged in a randomized order to minimize the possibility of response patterns.

A total of 224 participants were recruited between February and June 2017, with 56 students in every group. The subjects were chosen by simple random sampling and were asked to sign inform consents to use the questionnaires. Each subject filled the questionnaires through online forms, and the results attained from the questionnaires were statistically processed and analyzed. The frequency distribution method was applied to identify the degree of stigmatization from respondents. In addition, *chi-square* or *fisher* analysis was conducted to know the correlation between gender and attitudes towards psychiatry.

## Results

The questionnaire "Perception of Psychiatry" was distributed online to 250 medical students and alumni of Faculty of Medicine Universitas Indonesia. However, only 224 responders managed to fill the questionnaire, making the response rate 89.6 percent. The characteristic of the responders is presented in Table 1.

The participants were grouped into two categories: 1) medical students prior to psychiatry rotation, comprised the first- and fourth-year students; 2) after the exposure to psychiatry rotation, comprised the fifth-year students and fresh graduate alumni. The mean age of all subjects was 21.29 (SD = ±2.00).

Considering the perception of psychiatry as a discipline, more than 70 percent of respondents thought that psychiatry is scientific and intellectually challenging, a rapidly expanding frontier of medicine and a genuine and valid branch of medicine. However, about 40 percent of respondents agreed that psychiatrists make less money than other specialists.

The overall response showed a positive trend of their attitudes toward psychiatric treatment. Only about 10 percent of respondents agreed that psychiatric treatments are not evidence-based and psychiatrists could do very little for their patients. This positive response

**Table 1. Socio-demographic characteristic of respondents.**

| Variable(s) | Mean (SD) | N (n = 224) | % (n = 100%) |
|---|---|---|---|
| Age (years old) | 21.29 ± 2.00 | | |
| Gender | | | |
| Male | | 97 | 43.3 |
| Female | | 127 | 56.7 |
| Year of training | | | |
| Fresh-graduate | | 56 | 25.0 |
| 5$^{th}$-year | | 56 | 25.0 |
| 4$^{th}$-year | | 56 | 25.0 |
| 1$^{st}$-year | | 56 | 25.0 |

trend was also found in respondents' perception of psychiatrists. The responses from all subjects categorized by before and after psychiatric rotation are stated in Table 2.

Regarding perceptions of psychiatry as a career, the majority of the respondents disagreed that entering psychiatry is a waste of medical education. About 60 percent of respondents agreed that they would encourage a bright student to enter psychiatry. Nevertheless, nearly half of respondents agreed that psychiatry has low prestige among other medical disciplines. However, most respondents agreed that working with psychiatric patients is rewarding, and psychiatric illnesses deserve at least as much attention as physical illnesses. In addition, more than 80 percent of respondents agreed that psychiatric training in the university is of the highest quality, valuable, well-structured, informative, and mandatory.

Perceptions of psychiatry as a discipline throughout years of educations were primarily positive, as presented in Table 2. Throughout years of education, most of the respondents showed a positive perception of psychiatric treatments depicted from the first five questions of the section. The majority of the respondents also showed positive perceptions of psychiatrists as a role models in each group.

Regarding psychiatry as a career, there was a declining trend in the numbers of students that agreed to the statement "many students at this medical school are interested in pursuing psychiatry as a career" among the first, fourth, and fifth-year students. On the contrary, the number was shown to incline in the alumni group. In terms of perception of psychiatric training, most respondents showed a positive response.

Most respondents agreed that working with psychiatric patients is rewarding, and psychiatric illnesses deserve at least as much attention as physical illnesses. However, more than 70% of respondents agreed that psychiatric patients are emotionally draining and violent and unpredictable. Nevertheless, most respondents showed a positive perception of psychiatric training throughout all years of education.

According to gender, there were only two statistically significant statements in perceptions of psychiatry as a discipline. The responses are shown in Table 3. More than 60 percent of respondents in each group agreed that psychiatry is a rapidly expanding frontier of medicine. However, more female respondents agree with over 80 percent response, making a statistically significant difference with a $p$-value of 0.024. However, the proportion of respondents who agreed that psychiatrists make less money was higher in male group with a $p$-value of 0.000.

Regarding perceptions of psychiatry as a career, male and female respondents showed significant differences in almost all statements, except "students are attracted to psychiatry due to personal problems", whereby more than 60% of all respondents agreed. On the other hand, more than 95% of respondents disagreed with "entering psychiatry is a waste of medical education". More female students and alumni agreed that they would encourage bright students to

**Table 2. Perception of medical students toward psychiatry.**

| Variable | All Medical Students | | Pre-psychiatry rotation group | | Post-psychiatry rotation group | |
|---|---|---|---|---|---|---|
| | n = 224 | | n = 112 | | n = 112 | |
| | Agreed | Disagreed | Agreed | Disagreed | Agreed | Disagreed |
| | (n (%)) | (n (%)) | (n (%)) | (n (%)) | (n (%)) | (n (%)) |
| **Psychiatry as a discipline:** | | | | | | |
| Psychiatry is unscientific | 13 | 211 | 8 | 104 | 5 | 107 |
| | (6%) | (94%) | (7%) | (93%) | (4%) | (96%) |
| Psychiatry is a rapidly expanding frontier of medicine | 169 | 54 | 90 | 22 | 79 | 33 |
| | (75%) | (25%) | (80%) | (20%) | (71%) | (29%) |
| Psychiatry is intellectually challenging | 207 | 17 | 104 | 8 | 103 | 9 |
| | (92%) | (8%) | (93%) | (7%) | (92%) | (8%) |
| Psychiatry is not a genuine and valid branch of medicine. | 14 | 210 | 7 | 105 | 7 | 105 |
| | (6%) | (94%) | (6%) | (94%) | (6%) | (94%) |
| On average, psychiatrists make less money than other specialists. | 98 | 126 | 45 | 67 | 53 | 59 |
| | (44%) | (56%) | (40%) | (60%) | (47%) | (53%) |
| **Psychiatric treatments:** | | | | | | |
| Psychiatric treatments are not evidence-based. | 8 | 216 | 5 | 107 | 3 | 109 |
| | (4%) | (96%) | (4%) | (96%) | (3%) | (97%) |
| Psychiatric treatments are as effective as treatments in other branches of medicine. | 186 | 38 | 92 | 20 | 94 | 18 |
| | (83%) | (17%) | (82%) | (18%) | (84%) | (16%) |
| Psychiatric patients should be treated in specialized facilities. | 181 | 43 | 99 | 13 | 82 | 30 |
| | (81%) | (19%() | (88%) | (12%) | (73%) | (27%) |
| Most people who receive psychiatric treatment find it helpful. | 217 | 7 | 107 | 5 | 110 | 2 |
| | (97%) | (3%) | (96%) | (4%) | (98%) | (2%) |
| There is very little that psychiatrists can do for their patients. | 23 | 201 | 10 | 102 | 13 | 99 |
| | (10%) | (90%) | (9%) | (91%) | (12%) | (88%) |
| Psychiatric hospitals are little more than prisons. | 57 | 167 | 35 | 77 | 22 | 90 |
| | (25%) | (75%) | (31%) | (69%) | (20%) | (80%) |
| Psychiatrists have too much power over their patients. | 50 | 174 | 30 | 82 | 20 | 92 |
| | (22%) | (78%) | (27%) | (73%) | (18%) | (82%) |
| **Psychiatrists as role models:** | | | | | | |
| Most psychiatrists are not clear, logical thinkers. | 13 | 211 | 5 | 107 | 8 | 104 |
| | (6%) | (94%) | (4%) | (96%) | (7%) | (93%) |
| Psychiatrists are not good role models for medical students. | 8 | 216 | 5 | 107 | 3 | 109 |
| | (4%) | (96%) | (4%) | (96%) | (3%) | (97%) |
| Psychiatrists are difficult to talk to. | 20 | 204 | 11 | 101 | 9 | 103 |
| | (9%) | (91%) | (10%) | (90%) | (8%) | (92%) |
| Psychiatrists are not attentive enough to physiology. | 27 | 197 | 14 | 98 | 13 | 99 |
| | (12%) | (88%) | (13%) | (88%) | (12%) | (88%) |
| Psychiatry is filled with people whose medical skills are of low quality. | 7 | 217 | 3 | 109 | 4 | 108 |
| | (3%) | (97%) | (3%) | (97%) | (4%) | (96%) |
| **Psychiatry as a career:** | | | | | | |
| I would encourage a bright student to enter psychiatry. | 147 | 77 | 70 | 42 | 77 | 56 |
| | (66%) | (34%) | (63%) | (38%) | (69%) | (50%) |
| Psychiatry has low prestige among other medical disciplines. | 99 | 125 | 44 | 68 | 55 | 57 |
| | (44%) | (56%) | (39%) | (61%) | (49%) | (51%) |

(*Continued*)

**Table 2.** (Continued)

| Variable | All Medical Students | | Pre-psychiatry rotation group | | Post-psychiatry rotation group | |
|---|---|---|---|---|---|---|
| | n = 224 | | n = 112 | | n = 112 | |
| | Agreed | Disagreed | Agreed | Disagreed | Agreed | Disagreed |
| | (n (%)) | (n (%)) | (n (%)) | (n (%)) | (n (%)) | (n (%)) |
| Many students at this medical school are interested in pursuing psychiatry as a career. | 54 | 170 | 35 | 77 | 19 | 93 |
| | (24%) | (76%) | (31%) | (69%) | (17%) | (83%) |
| Students who could not obtain a residency position in other specialties eventually enter psychiatry. | 33 | 191 | 20 | 92 | 13 | 99 |
| | (15%) | (85%) | (18%) | (82%) | (12%) | (88%) |
| Students are generally attracted to psychiatry because of their own personal problems. | 143 | 81 | 71 | 41 | 72 | 40 |
| | (64%) | (36%) | (63%) | (37%) | (64%) | (36%) |
| My colleagues generally speak well of psychiatry. | 157 | 67 | 76 | 36 | 81 | 31 |
| | (70%) | (30%) | (68%) | (32%) | (72%) | (28%) |
| Entering psychiatry is a waste of medical education. | 5 | 219 | 2 | 110 | 3 | 109 |
| | (2%) | (98%) | (2%) | (98%) | (3%) | (97%) |
| **Psychiatric patients:** | | | | | | |
| Working with psychiatric patients is rewarding. | 193 | 31 | 96 | 16 | 97 | 15 |
| | (86%) | (14%) | (86%) | (14%) | (87%) | (13%) |
| Psychiatric patients are emotionally draining. | 182 | 42 | 94 | 18 | 88 | 24 |
| | (81%) | (19%) | (84%) | (16%) | (79%) | (21%) |
| Psychiatric patients tend to be violent and unpredictable. | 158 | 66 | 84 | 28 | 74 | 38 |
| | (70%) | (30%) | (75%) | (25%) | (66%) | (34%) |
| Psychiatric patients are highly appreciative of the care they receive. | 175 | 49 | 86 | 26 | 89 | 23 |
| | (78%) | (22%) | (77%) | (23%) | (79%) | (21%) |
| Psychiatric patients should be treated in specialized facilities away from general hospitals. | 108 | 116 | 71 | 41 | 37 | 75 |
| | (48%) | (52%) | (63%) | (37%) | (33%) | (67%) |
| Psychiatric patients are often more interesting to work with than other patients. | 121 | 103 | 62 | 50 | 59 | 53 |
| | (54%) | (46%) | (55%) | (45%) | (53%) | (47%) |
| Psychiatric illnesses deserve at least as much attention as physical illnesses. | 199 | 25 | 100 | 12 | 99 | 13 |
| | (88%) | (11%) | (89%) | (11%) | (88%) | (12%) |
| **Psychiatric training:** | | | | | | |
| Psychiatric teaching at this medical school is of the highest quality. | 187 | 37 | 86 | 26 | 101 | 11 |
| | (83%) | (17%) | (77%) | (23%) | (90%) | (10%) |
| Students at this medical school think that their psychiatric training has been valuable. | 197 | 27 | 93 | 19 | 104 | 8 |
| | (88%) | (12%) | (83%) | (17%) | (93%) | (7%) |
| Less time should be spent in the medical curriculum teaching psychiatry to medical students. | 38 | 203 | 18 | 94 | 20 | 92 |
| | (17%) | (83%) | (16%) | (84%) | (18%) | (82%) |
| Psychiatric rotations are well structured and informative. | 200 | 24 | 91 | 21 | 109 | 3 |
| | (89%) | (11%) | (81%) | (19%) | (97%) | (3%) |
| Psychiatry is so vague and imprecise that it cannot really be taught effectively. | 70 | 154 | 44 | 68 | 26 | 86 |
| | (31%) | (69%) | (39%) | (61%) | (23%) | (77%) |
| Psychiatric rotations should not be mandatory. | 25 | 199 | 17 | 95 | 8 | 104 |
| | (11%) | (89%) | (15%) | (85%) | (7%) | (93%) |

enter psychiatry, interested in pursuing psychiatry as a career, and their colleagues generally speak well of psychiatry. Meanwhile, male students agreed more to psychiatry, having lower prestige and those who could not attain residency in other specialties eventually chose psychiatry.

**Table 3. Perception of medical students toward psychiatry according to gender.**

| Variable | Male (n = 97) | | Female (n-127) | | p-value |
|---|---|---|---|---|---|
| | Agreed (n (%)) | Disagreed (n (%)) | Agreed (n (%)) | Disagreed (n (%)) | |
| **Perceptions of psychiatry as a discipline:** | | | | | |
| Psychiatry is unscientific | 8 (8.2%) | 89 (91.8%) | 5 (3.9%) | 122 (96.1%) | 0.172 |
| Psychiatry is a rapidly expanding frontier of medicine. | 66 (68.0%) | 31 (32.0%) | 103 (81.1%) | 24 (18.9%) | **0.024** |
| Psychiatry is intellectually challenging | 86 (88.7%) | 11 (11.3%) | 121 (95.3%) | 6 (4.7%) | 0.064 |
| Psychiatry is not a genuine and valid branch of medicine. | 6 (6.2%) | 91 (93.8%) | 8 (6.3%) | 119 (93.7%) | 0.972 |
| On average, psychiatrists make less money than other specialists. | 61 (62.9%) | 36 (37.1%) | 37 (29.1%) | 90 (70.9%) | **0.000** |
| **Perception of psychiatric treatments:** | | | | | |
| Psychiatric treatments are not evidence-based. | 4 (4.1%) | 93 (95.9%) | 4 (3.1%) | 123 (96.9%) | 0.697 |
| Psychiatric treatments are as effective as treatments in other branches of medicine. | 79 (81.4%) | 18 (18.6%) | 107 (84.3%) | 20 (15.7%) | 0.579 |
| Psychiatric patients should be treated in specialized facilities. | 73 (75.3%) | 24 (24.7%) | 108 (85.0%) | 19 (15.0%) | 0.065 |
| Most people who receive psychiatric treatment find it helpful. | 93 (95.9%) | 4 (4.1%) | 124 (97.6%) | 3 (2.4%) | 0.453 |
| There is very little that psychiatrists can do for their patients. | 9 (9.3%) | 88 (90.7%) | 14 (11.0%) | 113 (89.0%) | 0.670 |
| Psychiatric hospitals are little more than prisons. | 27 (27.8%) | 70 (72.2%) | 30 (23.6%) | 97 (76.4%) | 0.473 |
| Psychiatrists have too much power over their patients. | 24 (24.7%) | 73 (75.3%) | 26 (20.5%) | 101 (79.5%) | 0.447 |
| **Perceptions of psychiatrists as role models:** | | | | | |
| Most psychiatrists are not clear, logical thinkers. | 8 (8.2%) | 89 (91.8%) | 5 (3.9%) | 122 (96.1%) | 0.172 |
| Psychiatrists are not good role models for medical students. | 4 (4.'%) | 93 (95.9%) | 4 (3.1%) | 123 (96.9%) | 0.697 |
| Psychiatrists are difficult to talk to. | 11 (11.3%) | 86 (88.7%) | 9 (7.1%) | 118 (92.9%) | 0.269 |
| Psychiatrists are not attentive enough to physiology. | 16 (16.5%) | 81 (83.5%) | 11 (8.7%) | 116 (91.3%) | 0.074 |
| Psychiatry is filled with people whose medical skills are of low quality. | 5 (5.2%) | 92 (94.8%) | 2 (1.6%) | 125 (98.4%) | 0.127 |
| **Perceptions of psychiatry as a career:** | | | | | |
| I would encourage a bright student to enter psychiatry. | 54 (55.7%) | 43 (44.3%) | 93 (73.2%) | 34 (26.8%) | **0.006** |
| Psychiatry has low prestige among other medical disciplines. | 56 (57.7%) | 41 (42.3%) | 43 (33.9%) | 84 (66.1%) | **0.000** |
| Many students at this medical school are interested in pursuing psychiatry as a career. | 17 (17.5%) | 80 (82.5%) | 37 (29.1%) | 90 (70.9%) | **0.044** |
| Students who could not obtain a residency position in other specialties eventually enter psychiatry. | 21 (21.6%) | 76 (78.4%) | 12 (9.4%) | 115 (90.6%) | **0.011** |
| Students are generally attracted to psychiatry because of their own personal problems. | 63 (64.9%) | 34 (35.1%) | 80 (63.0%) | 47 (37.0%) | 0.763 |
| My colleagues generally speak well of psychiatry. | 60 (61.9%) | 37 (38.1%) | 97 (76.4%) | 30 (23.6%) | **0.019** |
| Entering psychiatry is a waste of medical education. | 3 (3.1%) | 94 (96.9%) | 2 (1.6%) | 125 (98.4%) | 0.446 |
| **Perceptions of psychiatric patients:** | | | | | |
| Working with psychiatric patients is rewarding. | 80 (82.5%) | 17 (17.5%) | 113 (89.0%) | 14 (11.0%) | 0.163 |
| Psychiatric patients are emotionally draining. | 81 (83.5%) | 16 (16.5%) | 101 (79.5%) | 26 (20.5%) | 0.450 |
| Psychiatric patients tend to be violent and unpredictable. | 62 (63.9%) | 35 (36.1%) | 96 (75.6%) | 31 (24.4%) | 0.058 |
| Psychiatric patients are highly appreciative of the care they receive. | 75 (77.3%) | 22 (22.7%) | 100 (78.7%) | 27 (21.3%) | 0.799 |
| Psychiatric patients should be treated in specialized facilities away from general hospitals. | 48 (49.5%) | 49 (50.5%) | 60 (47.2%) | 67 (52.8%) | 0.740 |
| Psychiatric patients are often more interesting to work with than other patients. | 51 (52.6%) | 46 (47.4%) | 70 (55.1%) | 57 (44.9%) | 0.705 |
| Psychiatric illnesses deserve at least as much attention as physical illnesses. | 83 (85.6%) | 14 (14.4%) | 116 (91.3%) | 11 (8.7%) | 0.174 |
| **Psychiatric training:** | | | | | |
| Psychiatric teaching at this medical school is of the highest quality. | 77 (79.4%) | 20 (20.6%) | 110 (86.6%) | 17 (13.4%) | 0.149 |
| Students at this medical school think that their psychiatric training has been valuable. | 86 (88.7%) | 11 (11.3%) | 111 (87.4%) | 16 (12.6%) | 0.774 |
| Less time should be spent in the medical curriculum teaching psychiatry to medical students. | 16 (16.5%) | 81 (83.5%) | 22 (17.3%) | 105 (82.7%) | 0.870 |
| Psychiatric rotations are well structured and informative. | 85 (87.6%) | 12 (12.4%) | 115 (90.6%) | 12 (9.4%) | 0.484 |
| Psychiatry is so vague and imprecise that it cannot really be taught effectively. | 34 (35.1%) | 63 (64.9%) | 36 (28.3%) | 91 (71.7%) | 0.283 |
| Psychiatric rotations should not be mandatory. | 16 (16.5%) | 81 (83.5%) | 9 (7.1%) | 118 (92.9%) | **0.027** |

Concerning medical students' attitudes toward psychiatric patients, data showed that gender did not affect most medical students' views. Only one statement that stated psychiatric rotation should not be mandatory has shown significant difference, with a $p$-value of 0.027 showing male group agreeing more to the statement. All the items in perception of psychiatric treatments, psychiatrists as role models, and psychiatric patients showed no statistically significant difference between gender and the views ($p>0.05$).

## Discussion

In Indonesia, medical education consists of three main stages for a student before practicing as a medical doctor or general practitioner independently: 1) three and half to four years of pre-clinical term; 2) two years of clinical post or clerkship; 3) one year of mandatory medical service from the Indonesian Ministry of Health after graduating as a Medical Doctor. We selected first- and fourth-year medical students whose desire and personal thought towards a future career path in medicine are undiluted and highly based on idealism, particularly with the naïve first-year students and the nearly determined fourth-year students. On the other hand, fifth-year students and fresh graduate alumni who have been exposed to psychiatric rotation were selected due to equipped experience with knowledge and practice to actual cases. Nevertheless, the freedom to experience other medical fields is still wide open [12].

This study assessed medical students' perceptions toward psychiatry as a discipline, psychiatric treatment, psychiatrists as role models, psychiatry as a career, psychiatric patients, and psychiatry training. While similar studies have been performed in several countries, none has been done in Indonesia. Therefore, we compared the results with other existing studies.

### Medical students' perspective towards discipline of psychiatry

Our findings suggested no significant difference in respondents' views toward the discipline of psychiatry throughout medical training. It corresponds with the findings from a study by Kuhnigk, et al. [13], although the further interpretation of the result was limited due to various students' characteristics, such as their personality and particular conditions they were in. However, many previous studies suggested favorable improvement of medical students' perspective towards psychiatry following psychiatric training [14–18].

Over the past years, psychiatry has been presumed as a last resort whenever the underlying cause of one patient's case has yet to be found. This presumption leads to an argument of psychiatry being unscientific or not a valid branch of medicine, as stated on the questionnaire. Despite the relevant and similar findings shown by earlier studies in Spain (2005) [9] and in India (2010) [19], Gulati, et al. revealed from their study that most of both interns and students did not consider psychiatry to be unscientific. The finding remarks that the development of psychiatry as a distinctive branch in medicine has inevitably risen and plays a vital role within our society [20].

Another critical finding implicated that about half of respondents, regardless of their exposure to psychiatric clerkship or length of study, perceived that psychiatrist had low wages compared to other specialties. The factor, as mentioned earlier, along with another financial drawback, such as lack of government funding, could affect the interest and attitude of medical students, regardless of the year of study, towards the discipline of psychiatry, as suggested by some previous studies [21].

There is also a significant difference between male and female students' perceptions of wages, whereas male students viewed psychiatrists as having lower incomes than other specialists. A previous study assessed the most common factors that could influence medical specialty decision making, found that anticipated income was one of the top factors for male students

[22]. Understanding that some male students might defer from studying psychiatry due to lower income is important, so future programs that attract students to enter psychiatry could explain the financial issue. A study in Ghana, another low-to-middle-income country, found that remuneration was essential in choosing psychiatry as a medical specialty. Over one-third of the students in the study would consider choosing psychiatry if they received financial incentives such as scholarships [23]. This financial incentive would be a topic that might be explored more in Indonesia since students going through medical residency in Indonesia must pay tuition fees and, therefore, could affect their decisions in choosing specialties.

Our findings revealed that most respondents from both groups found that psychiatry was intellectually challenging. Psychiatry incorporates medicine, psychology, sociology, and anthropology into the discipline. Therefore, emphasizing these aspects of psychiatry during clerkship or medical training through various previously mentioned measures could draw more interest towards the discipline, as suggested by Fischel et al. [24]. Identification and encouragement for students who showed significant interest in psychiatry had been shown to improve their attitudes and intention post-clerkship. Ay, Save, and Fidanoglu [25] stated that special training for psychiatric educators to address the problems encountered during medical education is needed and encouraged in medical institutions.

Medical students' views, skills, and knowledge at the end of clerkship have many implications regarding how well they will manage patients with mental illness issues later in their clinical practice. Thus, ensuring a good clerkship experience may indirectly affect the quality of service that mental illness patients would receive in the future [26].

There is also a possibility that the views towards psychiatry had already existed prior to the clinical psychiatry rotation, and it may or may not get improved after medical training [25,27]. One important note is that attitudes are determined by factors such as family, society, and media influence, all of which are affected by the ongoing stigma in general and are hardly modified [28].

## Medical students' perspective towards psychiatric treatment

Our study suggested the joyous prospect of psychiatric treatment circulated among the students, even before any formal education of psychiatric courses for the first-year students. The high positivity can be justified by recent discoveries in the modern era where the uncertain theories of psychiatric treatment in the previous years are evidence-based and scientifically proven. Skull drills and electrical shock therapy have evolved into a more holistic approach emphasizing community-based care with specific treatments according to the patient's needs and priorities [29].

Those findings are supported by the newly emerged discovery that proved a holistic approach to health and wellness, including physical and mental health. Furthermore, one study among medical students demonstrates that the stigma towards mental illness has started to decline [30]. Puspitasari, et al., from their study, mentioned that students who had the experience of visiting a psychologist or psychiatrist were proven to have better perception, knowledge, and attitude towards psychiatry and mental health disorders. At times, psychiatric treatment could be intriguing due to physicians' different approaches than other medical fields. However, the people exposed to the line of psychiatric treatment may have a better understanding and therefore have a decent attitude towards the field [31]. In addition, aside from the new psychiatric theories, the favourable responses from medical students may be due to their personal experience in psychiatric wards or with family members or friends who were previously treated due to mental illness prior to their enrolment in medical school. Over time, being involved in medical training, particularly for having hands-on experience with

psychiatric patients and their families, would make them practice empathetic psychiatric interviews and first-hand witness effective psychotherapy and psychopharmacology. These situations may result in fewer stigmatizing viewpoints.

Furthermore, studies indicated that being an undergraduate student in medicine may develop empathy and a sense of curiosity toward patients suffering from mental disorders. For example, Öster, et al., in their study, involved psychiatric patients mentioned that the participation of medical students in psychiatric care is essential. Not only did they learn about theory, but they also developed empathy from experience. However, this viewpoint would succeed if students would see the patients as individuals and subjects, not merely as learning objects [32].

Regarding the exposure of psychiatry, our study revealed there was not much significant difference in the perspective of psychiatric treatment between pre- and post-psychiatry rotation. This finding seemed to corroborate previous similar studies conducted in other countries. However, perspectives towards psychiatric treatment efficacy were varied [14,17,18]. Nevertheless, our results pointed out that more prolonged exposure to psychiatric wards or departments could still strengthen their knowledge concerning medical services provided by the psychiatrists.

Our analysis confirmed no notable differences between male and female respondents' views toward psychiatric treatment or psychotherapy. This outcome corresponds with previous studies that concluded that gender did not associate with contrast to mental illness [33]. However, the discovery differed from earlier studies that stated that females have a higher sympathetic view and positive incline towards mental illness [34]. Sarhan et al. [35] found that more male students had an optimistic view of psychiatry treatment efficacy. Nevertheless, the apparent reason behind the influence of gender is still debatable in many of the studies. Hence, this matter of subject requires more descriptive and elaborative data.

## Medical students' perspective toward psychiatry as a career

The overall attitude of respondents on psychiatry as a career was generally positive. This result is per previous studies [35,36]. Our study is also in line with previous studies, whereas women had a more positive attitude towards psychiatry as a career. This condition may be linked to a more profound social aspect in psychiatry than other medical specialties, in which women tend to have a higher interest, according to other studies [13,37]. In contrast, more male students viewed psychiatry as having low prestige and that those who could not obtain other residencies would then choose psychiatry.

Furthermore, only slightly over half of them would encourage bright students to choose psychiatry. A career preference in psychiatry is subjective to clinical rotation as there are attributable factors that could not be modified during the rotation [38]. Cutler, et al. [8] also suggested one of the factors in which the preference of medical students in selecting psychiatry was often influenced by the negative outlook their family members had on a psychiatrist. Such comment of "career in psychiatry is a waste of time" and the inability to recognize psychiatrist as a valid medical profession did exist.

From our study, the number of students who had a positive attitude after psychiatric rotation was increased. One possible reason for this might be the quality of psychiatric education and the opportunity to communicate with psychiatric teachers, patients, and families, allowing students to see a career in psychiatry as a potential and promising path [39]. Previous studies revealed the same result in which interest in pursuing psychiatry as a career increased after being exposed to psychiatry training [14–18].

Badmouthing by medical students and teachers could also be one of the contributing factors affecting decision-making to become a psychiatrist [40]. Psychiatry and family medicine,

general internal medicine, and general surgery frequently receive negative talk. How psychiatrist is perceived in medicine is essential because any negative prejudice could lead to a declining number of students taking psychiatry in the future. Despite the positive attitude from the students, a general acknowledgments that psychiatrist has low income and status in the eyes of society contributes to their decision in choosing psychiatry. Nevertheless, many students still assumed a psychiatrist as an intellectually challenging profession [10], which was in line with our data that most of the respondents agreed with the statement of encouraging bright students to enter psychiatry.

## Medical students' perspective toward psychiatrists as role models

Our study found that majority of the respondents had positive perceptions of psychiatrists as role models. Previous studies also noted positive outcomes that mentioned most students included in the study put respect on the skill and knowledge of psychiatrists [35,36,41]. This finding came out to be interesting, compared to different studies which had reported fewer positive responses towards psychiatrists [42]. The shifting trend of perspective towards psychiatrists as role models apparently will boost medical students' enthusiasm in pursuing a career as mental health professionals.

In our study, the positive perspective towards psychiatrists as role models was similar between pre- and post-psychiatry clinical rotation groups. Meanwhile, other studies revealed an increasing trend of positive attitudes after exposure to psychiatry [17,18]. This trend could be attributed to how the teachers and staff in the Department of Psychiatry presented themselves to clerkship students and how psychiatry was taught and delivered.

Moreover, the majority of the students from both male and female groups in our study had positive perspectives towards psychiatrists as role models, despite having no statistical significance. This result is similar to a previous study that showed a nearly proportional ratio of medical students who had positive perspectives towards psychiatrists as role models [35].

## Medical students' perspective towards psychiatric patients

Our study indicated that most of the respondents had positive attitudes towards psychiatric patients, which was also similar to previous studies [35,43]. Our finding could conclude that psychiatry rotation may contribute to the shifting trend of medical students' perspective towards psychiatric patients to be more positive, as supported by similar findings from previous studies [14,15,17,18]. Meanwhile, Aruna et al. [36] found an exciting result: final-year students were less sympathetic towards psychiatric patients than first-year students. Nevertheless, most of the students were optimistic regarding the treatability of psychiatric disorders.

In our study, the majority of the students of both genders had positive perspectives towards psychiatric patients. However, female students amounted to a slightly higher proportion, similar to a study by Sarhan, et al. [35]. This finding can be due to females' better empathy and understanding towards mental illness than males.

Another finding in our study revealed that the majority of the medical students agreed to consider psychiatric illness was as important as physical illness. Mental illness affects the brain and physical function; hence, the patient is incapable of making irrational judgments and is shown unacceptable behavior. In addition, the illness affects a patient's quality of life; therefore, medical intervention should be performed to improve current health conditions, as equal to physical illness.

Some students considered psychiatric patients violent, unpredictable, and emotionally draining regarding the behavior. According to a study by Naeem, et al., mood changes shown

by a patient could affect the perception of a healthcare professional, whereby doctors seemed to be emotionally draining to treat psychiatric patients with this condition [43].

Based on our study, the proportion of students' positive perspectives of working with psychiatric patients compared to other patients was generally in balance. Previous studies regarding medical students' attitudes toward psychiatric patients also revealed inconsistent results [17,35,44]. Throughout pre-clinical years, students gain knowledge and experience limited to written materials, tutors, and personal experience from the specialists. By the time students are exposed to psychiatric rotation, they tend to find that psychiatric patients in clinical settings have such complexity in appearance and personal engagement. However, medical students expressed various responses regarding this situation, to a certain extent that some found it fascinating. Acute psychiatric patients with behavioral issues, making them unpredictable, are considered dangerous. As a result, medical students, particularly freshmen, agree that psychiatric patients should be treated in specialized facilities [45]. However, the trend has been changing over time. Our data showed that the number of students who agreed with isolating psychiatric patients was decreased by half after the exposure to psychiatry rotation. Clinical rotation provides an opportunity for doctors to communicate directly with psychiatric patients. Thus, students would view doctors objectively, and therefore opposing viewpoints toward them would decrease [44].

## Medical students' perspective toward psychiatric training

Overall, most of the respondents in our study showed a positive perspective toward psychiatric training. The positive and negative perspective proportion was generally similar among pre- and post-psychiatric clinical rotations. Previous studies supported our findings that positive attitudes among students dominated the result. However, these studies suggested favorable improvement of a positive perspective towards psychiatry following the exposure to psychiatry in the clerkship years [14,16,17].

Our study revealed that most of the respondents thought that their psychiatric training had been valuable and should be mandatory. A similar result was also found in a study by Konwar et al. [17]. However, one-third of respondents in our study believed that psychiatry was vague and imprecise. It is predicted that several factors could enhance positive attitude towards psychiatric teaching and career, including a more structured curriculum, an increased number of psychiatric classes, better research in the biological underpinnings of various aspects of psychiatry, and a formal guideline towards the practice of psychiatry [46]. Upon medical school training, students who were initially considering psychiatry as a career had attended more lectures and psychiatric training [47]. We believe that the teaching method delivered, and the ambience created throughout the rotation play an essential role because the positive energy from the tutors would draw the attention and make students feel welcomed and appreciated.

In conclusion, the overall perception of medical students towards psychiatry as a discipline, psychiatric treatment, psychiatrists as role models, psychiatry as a career, psychiatric patients, and psychiatry training was positive. Correcting misapprehension and removing the stigma on psychiatry during medical education and clinical rotation can decrease negative perspectives on psychiatry. A direct experience and a substantial presence of role models can depict a more accurate and objective presentation of psychiatry to maintain and increase positive viewpoints among medical students towards psychiatric patients. In contrast, a social campaign could be an example for societies to perform.

For further development, a cohort study might be considered to identify factors affecting medical students' perceptions over a certain period. We suggest conducting a more extensive study scale, following this pilot study, involving different teaching sites and institutions to

obtain more data and collect broader perspectives. Although this university is the largest and oldest university in Indonesia, located in the capital city and students from this university represent diverse students from different places all over Indonesia. A qualitative study exploring the motives and reasons for the differences between male and female students' responses would also enrich further development of programs that aim to increase medical students' positive attitudes toward psychiatry as a discipline or a career choice.

## Acknowledgments

This study was conducted at the Faculty of Medicine Universitas Indonesia. The researchers thank all the medical students and recent graduates willing to participate in this study. We would also like to thank Prof. Norman Sartorius for his encouragement to study students' attitudes toward psychiatry in Indonesia.

## Author Contributions

**Conceptualization:** Fransiska Kaligis, Ribka Hillary, Nabilla Merdika Putri Kusuma, Helisa Rachel Patricie Sianipar, Camilla Sophi Ramadhanti.

**Data curation:** Fransiska Kaligis, Ribka Hillary, Nabilla Merdika Putri Kusuma, Helisa Rachel Patricie Sianipar, Camilla Sophi Ramadhanti.

**Formal analysis:** Fransiska Kaligis, Ribka Hillary, Nabilla Merdika Putri Kusuma, Helisa Rachel Patricie Sianipar, Camilla Sophi Ramadhanti, Ardi Findyartini, Madhyra Tri Indraswari, Clarissa Cita Magdalena, Garda Widhi Nurraga.

**Methodology:** Fransiska Kaligis, Ribka Hillary, Nabilla Merdika Putri Kusuma, Helisa Rachel Patricie Sianipar, Camilla Sophi Ramadhanti, Ardi Findyartini.

**Project administration:** Fransiska Kaligis.

**Validation:** Fransiska Kaligis.

**Writing – original draft:** Fransiska Kaligis, Ribka Hillary, Nabilla Merdika Putri Kusuma, Helisa Rachel Patricie Sianipar, Camilla Sophi Ramadhanti, Ardi Findyartini, Madhyra Tri Indraswari.

**Writing – review & editing:** Fransiska Kaligis, Ribka Hillary, Nabilla Merdika Putri Kusuma, Helisa Rachel Patricie Sianipar, Ardi Findyartini, Madhyra Tri Indraswari, Clarissa Cita Magdalena, Garda Widhi Nurraga.

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
