## [Decision Letter · Decision Letter 0]

26 Oct 2021

PONE-D-21-18251Medical students’ attitudes toward psychiatry in Indonesia: a pilot studyPLOS ONE

Dear Dr. Kaligis,

Thank you for submitting your manuscript to PLOS ONE. After careful consideration, we feel that it has merit but does not fully meet PLOS ONE’s publication criteria as it currently stands. Therefore, we invite you to submit a revised version of the manuscript that addresses the points raised during the review process.

Please, address the comments made by both reviewers.

We look forward to receiving your revised manuscript.

Kind regards,

Associate Professor Dr Muhammad Aziz Rahman,

MBBS, MPH, CertGTC, GCHECTL, PhD

Academic Editor

PLOS ONE

Reviewers' comments:

Reviewer's Responses to Questions

**Comments to the Author**

1. Is the manuscript technically sound, and do the data support the conclusions?

Reviewer #1: Partly

Reviewer #2: Partly

2. Has the statistical analysis been performed appropriately and rigorously? 

Reviewer #1: Yes

Reviewer #2: No

3. Have the authors made all data underlying the findings in their manuscript fully available?

Reviewer #1: Yes

Reviewer #2: Yes

4. Is the manuscript presented in an intelligible fashion and written in standard English?

Reviewer #1: No

Reviewer #2: No

5. Review Comments to the Author

Reviewer #1: 1. Conclusion part is missing

2. English language proficiency needs to be upgraded.

3. In the Discussion Part: I think that the whole discussion part may be written under a single point 'Discussion' with different paragraphs.

Reviewer #2: Title: Medical students’ attitude toward psychiatry in Indonesia: a pilot study

Reviewer’s comments:

TITLE: This is not clear why this is being called “a pilot study”. I have not found any explanation of this while reading through the paper

ABSTRACT: The result segment of the abstract is poorly written. There has been some repetitions and the results were not succinctly present.

INTRODUCTION:

• Several statements were identified that were not provided with in-text citations. The authors should revisit the introduction. For example, line 59-61, page 3 states, “Factors influencing this stigma among the society may be related to the lack of knowledge and information regarding the course of mental illness and its treatment” does not have any citation.

• Some portions of the introduction require rephrasing. For instance, line 72-75, page 4 states, “The media, culture, personality, and even history played part in the formed images of psychiatry. Another factor is the influences of training for medical students. Doctors usually have criticisms of how psychiatry lacks on science and evidence-based theory, as well as demands too much emotion, and criticism of how psychiatrists are not apt to be role models.” This seems unclear and complicated. The authors should try to make it easily understandable. Another sentence in the same page (line 81-82) is ambiguous.

MATERIALS AND METHODS:

• This is not clear why 2nd and 3rd year students were excluded

• The data collection process was not explained well

• There was no section that explained the sample size of the study

RESULTS:

• The socio-demographic characteristics seem incomplete

• In line 128-129 this was mentioned that the mean age of both categories was 21.29 with the same standard deviation. I am not sure whether this is correct or not

• Table 2 and Table 3 could be merged, and again to limit the table size the pre-psychiatry rotation group and post-psychiatry rotation groups can be analysed.

• The differences between pre-rotation and post-rotation groups could be provided as well

• In some cases, there was repetition of findings in narratives and table. This should be avoided

DISCUSSION:

• Discussion is well written

OVERALL COMMENTS:

• This paper is interesting but needs quite a lot of revision. The authors should think how the data could be presented in a more interesting way.

6. PLOS authors have the option to publish the peer review history of their article (what does this mean?). If published, this will include your full peer review and any attached files.

Reviewer #1: **Yes: **AHMK BakiBillah

Reviewer #2: No

---

## [Author Response · Author response to Decision Letter 0]

9 Dec 2021

Reviewer’s comments:

TITLE: This is not clear why this is being called “a pilot study”. I have not found any explanation of this while reading through the paper

In Indonesia itself, no research has been conducted about how the negative stigma is circulating and how it affects medical students' future decisions and preferences. The authors assumed it was appropriate to say pilot study as the first conducted study in Indonesia. However, since it has been done in other countries, the authors decided to remove the title “a pilot study”.

ABSTRACT: The result segment of the abstract is poorly written. There has been some repetitions and the results were not succinctly present.

A mix of positive and negative perceptions towards psychiatry was identified. The overall response was favorable to both before and after psychiatric rotation groups. We have modified the abstract to clarify and highlight the significant result. 

INTRODUCTION: 

• Several statements were identified that were not provided with in-text citations. The authors should revisit the introduction. For example, line 59-61, page 3 states, “Factors influencing this stigma among the society may be related to the lack of knowledge and information regarding the course of mental illness and its treatment” does not have any citation. 

Factors influencing this stigma in society may be related to the lack of knowledge and information regarding the course of mental illness and its treatment [3].

• Some portions of the introduction require rephrasing. For instance, line 72-75, page 4 states, “The media, culture, personality, and even history played part in the formed images of psychiatry. Another factor is the influences of training for medical students. Doctors usually have criticisms of how psychiatry lacks on science and evidence-based theory, as well as demands too much emotion, and criticism of how psychiatrists are not apt to be role models.” This seems unclear and complicated. The authors should try to make it easily understandable. Another sentence in the same page (line 81-82) is ambiguous.

The public's perspectives toward psychiatry have been primarily formed by the portrayal in the media, culture, and history. These images have also influenced how medical students are being trained. For example, it was found that doctors would criticize how psychiatry lacks evidence-based theories and demands emotions more than scientific understanding, as well as criticism of how psychiatrists are not apt to be role models [7,8]. 

Medical students were reported to have minimal knowledge in psychiatry with large amount of stigma prior to being exposed to psychiatric patients [8]. However, the same study also suggested that the level of stigma decreases as students enter psychiatry clinical rotation and meet psychiatric patients [8]. Another study in 2005 by Bulbena et al. [9] found that the percentage of students who were willing to choose psychiatry as their specialty increased to 10.4% after the exposure to psychiatric rotation. The factors contributed to these findings are not only subjective to the clinical rotation, but other factors unrelated to psychiatric training as well. Cutler et al. [10] showed that the preference of medical students in choosing psychiatry as a career was often affected by their own family members. Students who are surrounded by families with negative perspectives toward psychiatry would less likely choose psychiatry as their future career.

MATERIALS AND METHODS:

• This is not clear why 2nd and 3rd year students were excluded

We recruited students who can represents the 4 groups (before any psychiatric modules; after neuropsychiatry pre-clinical module but before clinical psychiatry rotation, after clinical psychiatry rotation, and fresh graduate of medical students). In Universitas Indonesia, pre-clinical neuro-psychiatry module is given in the last two months of the third year, whereas the psychiatry clinical rotation is done in the fifth year. Hence, we included the subject from: (1) first-year medical students before any exposure to psychiatry module, (2) fourth-year medical students who have passed pre-clinical psychiatry module but prior to clinical psychiatric rotation, (3) fifth-year medical students after the exposure to psychiatric rotation, (4) fresh graduate alumni.

• The data collection process was not explained well

A simple random sampling method was used to acquire samples. The subjects were randomly selected from a list of student ID numbers. The inclusion criteria were: (1) first-year medical students, (2) fourth-year medical students prior to psychiatric rotation, (3) fifth-year medical students after the exposure to psychiatric rotation, (4) fresh graduate alumni who were in the one-year mandatory medical service assignment (Internship).

A total of 224 participants were recruited between February and June 2017, with 56 students in every group. The subjects were chosen by simple random sampling and were asked to sign informed consent to use the questionnaires. Each subject filled the questionnaires through online forms, and the results attained from the questionnaires were statistically processed and analyzed.

• There was no section that explained the sample size of the study

A total of 224 participants were recruited between February and June 2017, with 56 students in every group.

RESULTS:

• The socio-demographic characteristics seem incomplete

The authors assume the socio-demographic characteristics are in accordance with the obtained data as per initial recruitment.

• In line 128-129 this was mentioned that the mean age of both categories was 21.29 with the same standard deviation. I am not sure whether this is correct or not

The mean age of 21.29 ± 2.00 refers to all subjects of 224 samples.

• Table 2 and Table 3 could be merged, and again to limit the table size the pre-psychiatry rotation group and post-psychiatry rotation groups can be analysed. 

Table 2 and Table 3 have been merged into new Table 2 (line 146). We also managed to differ pre- and post-psychiatry rotation in the Table 2.

• The differences between pre-rotation and post-rotation groups could be provided as well

Already written. Please refer to the new Table 2 in the article.

• In some cases, there was repetition of findings in narratives and table. This should be avoided

We already paraphrased the narrative findings not to be repetitive from the Table.

DISCUSSION:

• Discussion is well written

All parts, including the discussion, have been paraphrased for a better academic reading.

OVERALL COMMENTS:

• This paper is interesting but needs quite a lot of revision. The authors should think how the data could be presented in a more interesting way.

We already altered the data presentation, including the merged findings in the Table. Not to mention, we have checked the use of English to be more academic and avoid any ambiguity.

---

## [Decision Letter · Decision Letter 1]

7 Mar 2022

Medical students’ attitudes toward psychiatry in Indonesia

PONE-D-21-18251R1

Dear Dr. Kaligis,

We’re pleased to inform you that your manuscript has been judged scientifically suitable for publication and will be formally accepted for publication once it meets all outstanding technical requirements.

Kind regards,

Sonia Brito-Costa, Ph.D.

Academic Editor

PLOS ONE

Additional Editor Comments (optional):

Reviewers' comments:

Reviewer's Responses to Questions

**Comments to the Author**

1. If the authors have adequately addressed your comments raised in a previous round of review and you feel that this manuscript is now acceptable for publication, you may indicate that here to bypass the “Comments to the Author” section, enter your conflict of interest statement in the “Confidential to Editor” section, and submit your "Accept" recommendation.

Reviewer #2: All comments have been addressed

2. Is the manuscript technically sound, and do the data support the conclusions?

Reviewer #2: Yes

3. Has the statistical analysis been performed appropriately and rigorously? 

Reviewer #2: Yes

4. Have the authors made all data underlying the findings in their manuscript fully available?

Reviewer #2: Yes

5. Is the manuscript presented in an intelligible fashion and written in standard English?

Reviewer #2: Yes

6. Review Comments to the Author

Reviewer #2: (No Response)

7. PLOS authors have the option to publish the peer review history of their article (what does this mean?). If published, this will include your full peer review and any attached files.

Reviewer #2: No

---

## [Editor Report · Acceptance letter]

17 Mar 2022

PONE-D-21-18251R1 

Medical students’ attitudes toward psychiatry in Indonesia 

Dear Dr. Kaligis:

I'm pleased to inform you that your manuscript has been deemed suitable for publication in PLOS ONE. Congratulations! Your manuscript is now with our production department. 

Kind regards, 

on behalf of

Dr. Sonia Brito-Costa 

Academic Editor

PLOS ONE